# Dynamic Succession of Microbiota during Ensiling of Whole Plant Corn Following Inoculation with *Lactobacillus buchneri* and *Lactobacillus hilgardii* Alone or in Combination

**DOI:** 10.3390/microorganisms7120595

**Published:** 2019-11-21

**Authors:** Pascal Drouin, Julien Tremblay, Frédérique Chaucheyras-Durand

**Affiliations:** 1Lallemand Specialities Inc., Milwaukkee, WI 53218, USA; 2Energy, Mining and Environment Research Centre, National Research Council of Canada, Montréal, QC H4P 2R2, Canada; julien.tremblay@cnrc-nrc.gc.ca; 3Lallemand SAS, 19 rue des Briquetiers, 31702 Blagnac CEDEX, France; fchaucheyrasdurand@lallemand.com; 4Unité Mixte de Recherche 454 Microbiologie Environnement Digestif et Santé, Institut National de la Recherche Agronomique, Université Clermont Auvergne, 63000 Clermont-Ferrand, France

**Keywords:** silage, inoculant, lactic acid bacteria, microbial diversity, corn

## Abstract

Lactic acid bacteria (LAB) used as silage additives have been shown to improve several fermentation parameters, including aerobic stability. Inoculation with a combination of *Lactobacillus buchneri* NCIMB40788 and *Lactobacillus hilgardii* CNCM-I-4785, contributes to an increase in aerobic stability, compared to each strain inoculated independently. To understand the mode of action of the combination on the LAB community, a fermentation-kinetic study was performed on corn. Four treatments, Control, *Lb. buchneri,*
*Lb. hilgardii*, and a combination of the two strains, were fermented 1, 2, 4, 8, 16, 32, and 64 days. Corn silage inoculated by both strains had a lactate:acetate ratio of 0.59 after 64 days and a higher concentration of lactate than *Lb. buchneri*. Analysis of the microbiota by 16S and ITS amplicon metasequencing demonstrated that inoculation led to lower bacterial diversity after 1 day, from 129.4 down to 40.7 observed operational taxonomic units (OTUs). *Leuconostocaceae* represented the dominant population by day 1, with 48.1%. *Lactobacillaceae* dominated the succession by day 4, with 21.9%. After 32 days, inoculation by both strains had the lowest bacterial alpha diversity level, with 29.0 observed OTUs, compared to 61.3 for the Control. These results confirm the increased fermentation efficiency when the two *Lactobacillus* strains are co-inoculated, which also led to a specific yeast OTUs diversity profile, with *Hannaella* as the main OTU.

## 1. Introduction

Inoculation of silage, one of the main feed ingredients of dairy and beef cattle, has not only been credited with improving the quality of the preserved crops, but also with promoting several other benefits, including increasing animal performance [1], N and protein availability [2], reducing methane emissions [3], and adsorption of certain mycotoxins [4]. A recent meta-analysis of the impacts of homofermentative and facultative heterofermentative lactic acid bacteria (LAB) inoculants [5] confirmed these results and proposed some limitations and benefits to be gained by inoculation.

The succession of populations of lactic acid bacteria at the different phases of ensiling is a dynamic process that varies throughout the fermentation period, and high throughput sequencing technologies provide useful information on microbial shifts. Lin et al. [6] characterized more than 3400 bacterial isolates in a study using alfalfa and corn crops and the corresponding silage. Of those colonies, 306 were classified as homofermentative lactic acid bacteria (LAB). In corn, LAB succession was mainly dominated by *Lb. plantarum,* followed with *Ped. pentosaceus* [6]. These results provided a first glimpse into microbial population dynamics, but were limited by phylogenic identification of the different strains. Strains belonging to the genus *Streptococcus*, *Lactococcus*, and *Leuconostoc* were also identified in silage [7]. More recently, Brusetti et al. [8] studied the microbial succession of corn silage using the length heterogeneity PCR technique. After 1, 2, 6, 13, 20, and 30 days of fermentation, they observed a population shift from *Pediococcus* and *Weissella* on the first days of fermentation to *Enterobacter* as the dominant species after 30 days.

Inoculation of the forage with LAB at ensiling generally improves the overall quality of silage following their capacity to ferment soluble sugars to lactic acid along the homofermentative pathway, or to acetic acid and ethanol using heterofermentative pathways. Other molecules, including 1,2-propanediol are synthesized by certain species, the obligate heterofermentative *Lactobacillus buchneri* for example, and they improve aerobic stability [9]. Eikmeyer et al. [10] observed to what extent the inoculation of grass silage modified the LAB population after 14 and 58 days of fermentation. Without LAB inoculation, three genera, *Lactobacillus*, *Lactococcus*, and *Weissella*, dominated the silage. Inoculation with *Lb. buchneri* increased the size of the *Lactobacillus* genus in the treated samples after 58 days of storage.

Ensiling trials using *Lactobacillus hilgardii*, an obligate heterofermentative bacterium isolated from sugar cane and belonging to the same taxonomic group as *Lb. buchneri*, were recently published [11,12]. Some strains were not only effective in increasing the fermentation and aerobic stability of sugar cane, but also in improving the feed intake and milk yield of dairy cows [13,14]. Commercial microbial silage additive formulations often combined different strains of LAB with contrasting functions, for example a fast fermenting strain and one with increasing aerobic stability, but few microbial diversity studies have tried to understand the successional patterns following inoculation with multiple strains.

The objective of this study was thus to investigate microbial succession during the ensiling process by performing a kinetic analysis of whole-plant corn silage with up to 64 days of incubation in controlled conditions. Our second objective was to understand how whole plant corn inoculated with one of two obligate heterofermentative LAB strains, or a combination of the two, influences microbial succession patterns, determined by metasequencing of 16S rDNA and ITS (internal transcribed spacer) amplicon and analyzing their corresponding fermentation profiles using analytic chemistry.

## 2. Materials and Methods

### 2.1. Ensiling and Experimental Treatments

Whole-plant corn forage (*Zea mays* L. cultivar P1498HR) fertilized with 2 metric tons per ha of manure, 18 kg per ha of 10-20-20 starter fertilizer applied prior to planting, and 14 kg per ha of nitrogen during the growing season, was harvested on 2 October 2015 at the kernel half milk line and at a mean dry matter content of 352.0 g DM·kg^−1^ (*pF* = 0.507). No fungicides were used on the corn. The maximum temperature on the two days prior to harvesting was 14 and 13 °C, and the minimum was 4 °C. The self-propelled forage harvester (John Deere 8800) was set to chop at a theoretical length of 12.7 mm and was equipped with a KernelStar processor unit (John Deere). At harvesting, Acid Detergent Fiber (ADF) was at 21.2, Neutral Detergent Fiber (NDF) was at 36.6, ethanol-soluble carbohydrates were at 1.8, and crude proteins were at 6.1 % DM. After coarse mixing, the chopped corn was divided into four 30 kg piles prior to application of the treatments by alternatively spraying small volumes of inoculant and mixing, and was completed by mixing thoroughly. Initial counts of lactic acid bacteria performed on De Man, Rogosa, Sharpe (MRS) plates supplemented with cycloheximide (100 μg·L^−1^) was of 5.81 log_10_ CFU·g^−1^ fresh forage (ff). The treatments consisted of well water (Control), *Lactobacillus buchneri* NCIMB 40788 (LB) (4 × 10^5^ CFU·g^−1^ fresh forage), *Lactobacillus hilgardii* CNCM-I-4785 (LH) (4 × 10^5^ CFU·g^−1^ fresh forage, and a combination of the two *Lactobacillus* strains (NCIMB 40788 and CNCM-I-4785-LB-LH) at an equal ratio (2 × 10^5^ CFU·g^−1^ fresh forage, each strain). After application of the corresponding treatments, plastic bags (20 × 30 cm, 2-ply 3 mil polyethylene, Western Brands LLC, Ohio, OH, USA) were filled with 400 g of forage and immediately vacuumed and sealed on a commercial vacuum sealer (Model 3000, Weston Brands LLC). The mini-silos were stored at ambient temperature (~21 °C) in a storage room in the dark. They were disposed according to a complete randomized pattern and to allow good air-flow to insure stable temperature. In the week prior to inoculation, a cell-count was performed on the inoculant to the adjust application rate on MRS plates (Oxoid–Thermo Scientific, Hampshire, UK).

A total of 160 vacuum-bag mini-silos were prepared, 40 per treatment, with five repetitions for each treatment × time factors. The mini-silos were incubated using a completely randomized design (4 inoculation treatments × 8 opening periods). Mini-silos were opened after 0, 1, 2, 4, 8, 16, 32, and 64 days of fermentation. After the given incubation time, the corresponding mini-silos were opened, mixed, and three subsamples (fermentation profile, DNA extraction, metabolites extraction) from individual silos were immediately frozen in liquid nitrogen and stored at −80 °C.

### 2.2. Fermentation Profiles and Physical Characterization

Volatile fatty acids (VFA), lactic acid, ethanol, and pH were analyzed at Cumberland Valley Analytical Services. Briefly, each fermented silage sample was mixed, and a 25 g wet sample was taken and diluted with 200 mL deionized water. After overnight incubation under refrigeration, the sample mixture was blended for two minutes and filtered through coarse (20–25 μm particle retention) filter paper. For volatile fatty acid (VFA) quantification, 3 mL of the extract was filtered through a 0.2 μm filter membrane and a 1.0 μL subsample was injected into a Perkin Elmer AutoSystem Gas Chromatograph (Perkin Elmer, Waltham, MA, USA) equipped with a Restek column packed with Stabilwax-DA (Restek, Bellefonte, PA, USA). For lactic acid quantification, a 1:1 ratio of extract to deionized water was placed in a YSI 2700 Select Biochemistry Analyzer (YSI, Yellow Springs, OH, USA) to determine l-lactic acid. The pH was read prior to analysis of the titratable acidity from 30 mL of the previous extract by a Mettler DL12 Titrator (Mettler-Toledo, Columbus, OH, USA) using 0.1 N NaOH, pH 6.5.

### 2.3. DNA Extraction

The DNA extraction and purification protocol were adapted from the methodology proposed by Romero et al. [15] and Zhou et al. [16]. DNA extraction was performed from repetition 1, 2, and 3. Five grams of silage samples were weighed in a 50 mL conical centrifugation tube and suspended with 10 mL of sterile deionized water. The samples were then sonicated (Branson model 8800 ultrasonic water bath, 40 kHz) for five minutes and vortexed for one minute. A 3 mL aliquot of the corn silage suspension was centrifuged, and the pellet was transferred to tubes containing beads of the PowerLyzer Soil DNA Isolation Kit (MoBio Laboratories, Carlsbad, NM, USA). Microbial lysis was optimized by two minutes of mechanical lysis in a MixerMill 400 (Retsch, Inc., Haan, Germany) at a speed of 15 cycles per second. DNA isolation then proceeded according to the manufacturer’s protocol. The concentration of the DNA was measured on a spectrophotometer (Nanodrop Technology, Cambridge, UK) and quality was measured by agarose gel electrophoresis (1% agarose). The concentration of DNA was standardized for all samples at 2 ng·μL^−1^.

### 2.4. High-Throughput Sequencing and Bioinformatics Analysis

For amplicon sequencing, the libraries were prepared according to the Illumina 16S Metagenomic Sequencing Library Preparation guide (Part # 15044223 Rev. B), except that a Qiagen HotStar MasterMix (Toronto, ON, Canada) was used for the first PCR (amplicon PCR) and half the volume of reagents was used for the second PCR (index PCR). This protocol includes a PCR cleanup step that uses AMPure XP beads to purify amplicons from free primers and primer dimers. The template specific primers were as follows (without the overhang adapter sequence): 515F (5′-GTGCCAGCMGCCGCGGTAA-3′) and 806R (5′-GGACTACHVGGGTWTCTAAT-3′) for 16S amplification from the V4 hyper-variable region, and with the ITS region 1 specific primers ITS1F (5′-CTTGGTCATTTAGAGGAAGTAA-3′) and 58A2R (5′-CTGCGTTCTTCATCGAT-3′) for the amplification of fungi [17]. The amplicon PCR reaction was carried out for 30 cycles with annealing temperatures of 55 °C for 16S and 45 °C for ITS. Diluted pooled samples were loaded on an Illumina MiSeq and sequenced using a 500-cycle MiSeq Reagent Kit v2 (San Diego, CA, USA, adapted from Yergeau et al., [17]). The average size of the amplicon sequences were 293 bp and 276 bp for the 16S and ITS regions, respectively.

Sequences were analyzed as described by Tremblay et al. [18]. Briefly, common sequence contaminants (i.e., Illumina adapters and PhiX spike-in reads) were removed from raw sequences using a kmer matching tool (DUK; http://duk.sourceforge.net/). Filtered reads were assembled with the FLASH software version 1.2.11 [19]. The assembled amplicons were trimmed to remove forward and reverse primer sequences in each read using in-house Perl scripts. Paired-end assembled amplicons were then filtered for quality. Sequences with more than one undefined nucleotide, an average Phred score lower than 30, or more than 10 nucleotides with a quality score of less than 10 were discarded. De novo operational taxonomic unit (OTU) generation was performed using a three step-clustering pipeline. Briefly, quality-controlled sequences were dereplicated at 100% identity. These 100% identity clustered reads were then denoised at 99% identity using dnaclust [20]. Clusters of fewer than three reads were discarded, and the remaining clusters were scanned for chimeras using UCHIME de novo, followed by a UCHIME reference using the Broad’s Institute 16S rRNA Gold reference database. The remaining clusters were clustered at 97% identity (dnaclust) to generate OTU.

Taxonomy assignment of the resulting bacterial and archaeal OTUs was performed using the RDP classifier with a modified Greengenes training set built from a concatenation of the Greengenes database (version 13_5). Taxonomy assignment of fungal OTUs was performed using the SH Qiime release 20.11.2016 database (version 7.1). Hierarchical tree files were generated with in-house Perl scripts and used to generate training sets using the RDP classifier (v2.5) training set generator’s functionality [21]. A bootstrap cutoff of 50% was applied to each query to generate final taxonomic lineages. With taxonomic lineages in hand, OTU tables were generated and normalized using edgeR’s Relative Log Expression (RLE) method [22], as suggested by McMurdie and Holmes [23]. 

Bacterial/archaeal OTU representative sequences were aligned on a Greengenes core reference alignment [24]. Alignments were filtered to keep the V4 hypervariable region of the alignment. A phylogenetic tree was then built from this alignment with FastTree [25]. Alpha diversity (observed OTUs) was computed from un-normalized OTU tables, rarefied at 1000 reads with 10 permutations, for both 16S and ITS. For 16S data, weighted Unifrac distances were generated using the normalized OTU table and the phylogenetic tree previously generated. For ITS data, the Bray–Curtis dissimilarity metric was generated from the normalized OTU table. Taxonomic profiles were computed from the normalized OTU tables using the QIIME software suite version 2017.6.0 [26,27]. Permanova analyses with 1000 permutations were done in R using the adonis function of the vegan library. One sample from the 16S dataset (LB–LH, day 16 experimental condition) and one sample from the ITS dataset (control day 64 experimental condition) failed to give enough reads for downstream analyses and were therefore excluded from our analyses.

The 16S rDNA sequences from the microbiota analyses have been deposited in the NCBI BioProject database under the study accession number PRJNA510327.

### 2.5. Statistical Analysis

The effect of the incubation time on the fermentation parameters and the differences between the inoculation treatments were analyzed using a mixed-effect model with temporal pseudoreplication [28] using R version 3.3.3 [29]. The nmle package was used by the mixed-effect model [30]. The fixed-effect formula was the tested parameter–treatment, while the random-effects formula was ~day|silo. The following fermentation parameters were individually tested by the model: pH, lactic acid, acetic acid, and the ratio of lactic acid to the sum of acetic acid and ethanol (LA/AAEtOH). A linear model was used in the preparation of some figures using the equation y~ns(x^3^) from the splines package of R (version 3.4.0). The confidence interval boarding the trend lines was computed using the predictdf function of the package from *t*-based data and from standard error results of the glm function. The confidence interval was set to 95%. Significant differences were declared at *p* < 0.05 and as biological trends at *p* < 0.10.

In order to test the difference in OTUs between treatments at the Phylum, Order, Family, and Genus level, comparisons for each time period were performed independently using GLM (generalized linear model) with the treatment as independent variable and the targeted OTU as the response variable (e.g. Lactobacillaceae). 

According to our experimental design, differentially abundant OTUs (DAO) were assessed with edgeR (v3.10.2) using its GLM approach [31], with OTU table raw count matrices as inputs. OTUs having a logFC (log fold-change) ratio equal or higher than 1.5 and FDR (false discovery rate) <0.05 were considered as differentially abundant. Contrast analyses were performed between openings and the time of fermentation to generate statistical differences from the OTU sequences selected by the edgeR package version 3.16. The log transformation of the FC (fold change) and CPM (counts per million) values were used to generate histograms using in-house scripts, including ggplot functions. Lastly, both the 16S amplicon data and the ITS amplicon data were used to perform a linear discriminant analysis using the MASS package (v7.3-47) in R [32].

## 3. Results and Discussion

### 3.1. Eubacterial Succession over the Incubation Period

To date, only a few metagenomic studies of silages have been reported [10,15,33,34]. Our results showed that the populations of eubacteria observed on the corn samples collected immediately after inoculation, but prior to ensiling, were mainly composed of genera belonging to Proteobacteria (56.4 ± 1.5%) orders Pseudomonadales, Xanthomonadales, and Enterobacteriales, and Bacteriodetes (37.4 ± 1.7%) orders Sphingobacteriales and Flavobacteriales. Samples of the four treatments harbored identical relative abundance of these different taxonomic orders, with slight variations in Flavobacteriales in some samples (14.6 to 20.8%). The genera *Flavobacterium* and *Chryseobacterium,* belonging to the Flavobacteriales order, dominated the fresh forage samples with means of 10.2 ± 1.4% and 7.0 ± 0.6% of total OTUs, respectively. Sequences affiliated to the Lactobacillales order represented only 3.2 ± 0.9% of the total population in the fresh samples. Lactobacillales were substantial contributors of the Firmicutes phylum, with the *Leuconostocaceae* family representing between 60% and 100% of the fresh forage sample composition. Our results for fresh forage do not corroborate those of Gharechahi et al. in maize silage [35], but are more in line with the diversity observed in fresh small grains [33] or oats [15], as well as with data published by McGarvey et al. [36] on alfalfa. The Shannon diversity index was highest for the fresh maize samples, with a mean of 5.26 ± 0.27. Additional microbial community profiles from fresh corn samples taken one month prior to ensiling were consistent with our results for fresh forage (see Appendix A).

During the fermentation period, bacterial succession could be grouped in three phases (Figure 1). The first phase covered the first few days of fermentation with *Leuconostocaceae* as the dominant OTU. The second phase, from day 4 to 16, was defined by OTUs related to *Lactobacillaceae*. The third phase, from day 32, was characterized by the succession of heterofermentative *Lactobacillus* OTUs. Going forward, it is important to underscore that, starting from day 8, the bacterial community successions were largely dominated by OTUs assigned to the *Lactobacillaceae* family (in red in Figure 1A) and *Lactobacillus* genus (in blue in Figure 1A). Throughout the text, these two taxa will be referred to and represented as two distinct entities. This succession of lactic acid bacteria genera has already been described [7]. It has also been reported in different forage species [37,38], but the resolution offered by next generation sequencing provided insights into the slight differences between treatments following inoculation of similar taxa.

Within 1 to 2 days of incubation, a major shift in microbial community profiles was observed (Figure 1A), whatever the treatment. In that period, Firmicutes became dominant with 56–66% of the sequences, followed by Proteobacteria (31–44%, with *Enterobacteriaceae* with 30–41% of the sequences). The Phylum Bacteriodetes represented only 1.3 ± 0.2% (*p* = 0.384) of the reads after 2 days. Within Firmicutes, the families *Leuconostocaceae* and *Streptococcaceae* accounted for respectively 44–52% and 7–11% of the total population of the different silages. *Streptococcaceae* OTUs were related to the genus *Lactococcus.* Within the Proteobacteria phylum, the *Enterobacteriaceae* family was mainly composed of OTUs from the genera *Citrobacter, Brenneria, Enterobacter, Erwinia*, and *Serratia*. Our results after 1 day of incubation are in line with the diversity reported at time 0 by Gharechahi et al. [35] and to data after 1 and 2 days from Keshri et al. [39].

After two days of fermentation, the diversity profiles resembled those found after 1 day (Figure 1A). The main difference was the higher relative abundance of *Lactobacillaceae*, which increased from 2.6 ± 0.4% (Control) to 8.9 ± 1.4% (LB–LH) (*p* < 0.001), and *Lactobacillus*, in particular in LH and LB–LH treated silages, with abundance of 5.2 ± 1.1% and 7.5 ± 1.3%, respectively (*p* < 0.001 between the four treatments).

Samples incubated for 4, 8, and 16 days presented a different bacterial profile from those observed at 1 and 2 days. Figure 2 shows the microbial changes observed over time and was generated using the two dendrograms computed by correlation analysis from the heatmap (see Appendix A). At days 4, 8, and 16, a major increase in the relative abundance of *Lactobacillaceae* members was observed, with proportions ranging from 16.4 ± 0.8% (day 4, Control) to 56.1 ± 2.2% (day 16, LH). *Lactobacillus* still represented a small fraction of the community, but unassigned OTUs affiliated to *Lactobacillales* or *Lactobacillaceae* were more abundant. The *Enterobacteriaceae* genera *Citrobacter* and *Serratia* were still present, and the proportion of *Citrobacter* was often higher (day 16: 9.9% to 14.8%, *p* = 0.077 between treatments) than earlier in the ensiling process. *Enterobacteriaceae* were more abundant in LB treated samples (day 16: 33.0 ± 3.1%) than in LH treated samples (23.3 ± 2.8%). Relative abundance of OTUs related to *Leuconostocaceae* was lower during this phase of fermentation compared to the observed values at 1 and 2 days of incubation, possibly related to their poor tolerance of low pH [7]. LH and LB–LH treated silages had smaller proportions of this family, with 2.8 ± 0.3% and 2.3 ± 0.2%, respectively (*p* < 0.001 between treatments), compared to 4.0 ± 0.3% in the Control and 4.1 ± 0.2% in LB.

After 16 days of fermentation, the smallest proportion of *Lactobacillus* related OTUs was found in the Control silages (8.3 ± 1.2% in the Control versus 19.4 ± 1.8% (LH) to 25.3 ± 4.4% (LB–LH) in inoculated silages (*p* < 0.001)). OTUs affiliated to the *Pediococcus* genus were retrieved in this fermentation phase whatever the treatment (days 4 to 16). In the Control samples, *Lactococcus* slowly disappeared during this period, and by day 8, they represented only a minor fraction of the community. *Lactococcus* represented a higher proportion of the population in LB samples on day 4 (14.4 ± 1.3%, *p* < 0.001), and still represented a substantial proportion of the diversity in LB samples at day 8 (6.1 ± 1.9%, *p* = 0.009). Lastly, at day 16, OTUs linked to the genus *Brenneria* (*Enterobacteriaceae*) were found in all inoculated silage samples, with LB and LB–LH having the highest proportion (7.3 ± 1.2% to 9.0 ± 1.2%, respectively), and LH and the Control having the lowest (~5%) (*p* = 0.012).

In the third phase, the relative abundance of Lactobacillales increased markedly in the three treated silages (up to 92.3 ± 1.6%, 94.1 ± 0.2%, and 94.4 ± 0.7% in LB, LH, and LB–LH, respectively, by day 64). The increase was not observed in the Control silages, in which Lactobacillales related OTUs represented only 74.8 ± 2.4% (*p* < 0.001 between treatments). This result is in agreement with the observation made by Eikmeyer et al. [10] following microbiota analysis of treated and untreated grass silage and suggests the establishment of inoculated strains in the silage, or the promotion of other *Lactobacillus* species due to inoculation.

*Leuconostoc* was still present in the Control at the last opening, with 8.6 ± 1.2% of the relative abundance, along with unassigned *Leuconostocaceae* (12.9 ± 1.6% versus a mean of 1.0% in the treated samples, *p* < 0.001). *Citrobacter* (7.1 ± 0.6%)*, Brenneria* (4.0 ± 0.5%)*,* and *Erwinia* (2.2 ± 0.5%) accounted for another 13% of the population in the Control samples (Figure 1). The number of OTUs affiliated to these three genera was about ten times smaller in the treated samples (*p* < 0.001 for all three OTUs between treatments).

In a study performed using a similar experimental design on corn silage, with sampling time between 3 h to 90 days, Keshri et al. [39] observed similar phases of colonization and of bacterial richness evolution. They compared a non-inoculated control to silage, inoculated with a *L. plantarum* strain, but had the same interactions between *Leuconostocaceae* and *Lactobacillaceae* in relation with time of fermentation. Similar to their results, we observed that bacterial richness dropped after ensiling, then increased during the transition from *Leuconostocaceae* to the *Lactobacillaceae*, and then dropped again toward the transition to the heterofermentative LAB strains (Appendix A).

Principal component analysis was performed using the weighted unifrac distance values computed between the microbiota associated with the different types of silage and the course of fermentation (Figure 1B). The x-axis accounted for 67.6% of the observed differences and showed the shift in population between the three main periods, days 1 and 2, days 4 to 16, and days 32 to 64. Results from a PERMANOVA analysis show that the treatments, opening periods, and the interaction treatments:openings were significant, at *p* < 0.001, for the 16S amplicons. The y-axis separated the samples according to the duration of fermentation, which explained 19.3% of the changes in diversity between the samples. The Control samples (Figure 1B) clearly differed from the inoculated samples at 32 and 64 days of fermentation, indicating that the succession of the bacterial population slowed down after 16 days of fermentation. This observation means that inoculation favors a faster succession to a potentially more stable silage, with fewer active species.

In addition, a discriminant analysis (Figure 1C) was performed using the complete OTU dataset, after exclusion of the fresh samples. The results of the analysis showed that microbial succession followed similar trends in LB–LH and LH, but was more distinct between LB and Control. The comparison of the relative abundance of Lactobacillales could explain part of the observed difference (see Figure 3 and the heatmap Appendix A in the Appendix A). Considering that the discriminant analysis includes the effects of both time and treatment, understanding the links to a specific parameter is not easy.

To advance our understanding of the difference in microbial succession between treatments, variations in OTU abundance were subjected to contrast analysis using the edgeR package. Two values were generated for each individual OTU by pairwise comparison between treatments. The variation in the number of individual OTUs between two treatments was represented as logFC (fold change), and the difference in the number of individual OTUs was represented as logCPM (count per million). Although the computation was performed for each opening, only the results from the last opening are discussed here (Figure 3). To understand the dynamic for individual OTUs, the sequences were manually compared to the NCBI database in order to annotate the phylogenic identification. Additional analyses of the contrasts using this manual annotation are presented in the Appendix A in the form of two tables (Appendix A), comparing the identity of the main OTUs that increased or decreased between the different treatments. Taxonomic identification of one key OTU pointed to several species. The highest scoring hits against the sequence of this OTU (303 bp, % identity <99.26) were affiliated to either *Lb. hilgardii*, *Lb. diolivorans*, or *Lb. farraginis* (LH-LD-LF) following a BLAST search. After 64 days of fermentation, Control samples displayed higher population diversity than LB, LH, or LB–LH (Figure 3 and Appendix A). The OTU (253 bp, % identity 100.00) with the highest logFC value (−8.69) was assigned to *Bacillus pumilus* or *Bacillus safensis*. *B. pumilus* is often observed during aerobic deterioration [40]. This could mean that inoculation had an impact on the integrity of *Bacillus* spores, thereby potentially reducing the abundance of this specific OTU. Several OTUs related to Lactobacillales were more abundant in the Control samples, as well as OTUs related to Burkholderiales and Pseudomonadales. When we compared Control and LB microbiota (Figure 3), several OTUs affiliated as *Lactobacillus* were more abundant in LB, including *Lb. buchneri*, but also *Lb. ingluviei*, *Lb. pentosiphilus, Lb. harbinensis*, and *Lb. brevis*. Three OTUs related to LH-LD-LF were also observed, one of which may be related to *Lb. diolivorans*. The LH-LD-LF related OTU was also observed in LB–LH or LH at day 32, and at a lower intensity at day 64. Inoculation with LH either alone or combined with LB led to a reduced set of differentially abundant OTUs versus Control (three in LB–LH samples and two in LH samples). One of these OTUs was affiliated as either LH-LD-LF and we hypothesize that it was *Lb. hilgardii* with this treatment. For LB–LH, the three OTUs identified by the contrast analysis were LH-LD-LF (putatively *Lb. hilgardii*), *Lb. buchneri* (280 bp, % identify 100.00), and *Lb. helveticus* (251 bp, % identify 96.02). The two OTUs linked to LH were LH-LD-LF and *Lb. helveticus*. The OTU related to *Lb. brevis* (293 bp, % identify 99.63) was generally absent from the LB–LH and LH treatments, while OTU related to *Lb. helveticus* was absent from LB but present in LH. Few diversity studies on silages mention having identified *Lb. helveticus*, but a study of lactic acid bacteria diversity in tomato pomace silage reported that 4% of the isolates belonged to *Lb. helveticus* [41], which suggests this species is part of the common lactic bacteria community. However, direct inoculation of wilted rice straw with a *Lb. helveticus* strain had no effect on the ensiling process, and the strain only survived for five days post-inoculation [42]. When we compared LB and LB–LH, some of the previously discussed OTUs were more abundant in LB, including OTU related to *Lb. ingluviei* (252 bp, % identify 98.41), *Lb. kimchicus* (272 bp, % identify 99.62), *Lb. harbinensis* (251 bp, % identify 99.60), and *Lb. brevis* (293 pb, % identity 99.63). In a recent metasequencing study in which corn stover was inoculated with different *Lactobacillus* strains, *Lb. brevis* was identified during the early ensiling phase, i.e., between 10 and 30 days of fermentation, in the un-inoculated control and *Lb. brevis* or *Lb. parafarraginis* treated as forage only [43]. We observed similar trends, but after a slightly longer incubation period, after inoculation with LB. Two OTUs affiliated to the genus *Stenotrophomonas* were specific to LB. *Stenotrophomonas* are encountered in the plant rhizosphere and endosphere and *Stenotrophomonas* is a common soil microorganism [44].

### 3.2. Fungal Succession during the Course of the Incubation Period

Although major modifications of the fungal community were observed during fermentation, the fungal succession pattern was less distinct than that of bacteria (Figure 4A,B). Two main phases were observed, and an increase in the yeast population, with the final high abundance of *Candida, Dipodascus,* or *Hannaella*, were the main changes in this succession.

In the fresh samples, the majority of OTUs were affiliated to *Epicoccum*, *Cladosporium*, *Giberella, Bulleromyces*, *Neosetophoma*, and *Sarocladium* genera. The genus *Epicoccum* completely disappeared in the first day of ensiling, while *Neosetophoma* (1 day: ranging from 15.9 to 23.7%; 2 days: ranging from 15.1 to 17.8%) and *Peyronellaea* (1 day: 3.4 to 10.5%; 2 days: 12.7 to 22.1%) became the dominant genera after 1 and 2 days, respectively. *Epicoccum* is a ubiquitous saprophytic mould often associated with plant senescence and the early phases of composting [45]. *Neosetophoma* is an Ascomycete belonging to the Phoma group which may have saprophytic functions [46]. Several species of the genus *Peyronellaea* are plant pathogens. The *Sarocladium* were present throughout the fermentation period at abundance ranging from 1.0 ± 0.1% (Control, day 64) to 15.7 ± 2.5% (LB, day 2). Most *Sarocladium* species are important plant pathogens [47].

The second fungal succession phase extended from day 4 to day 32 and corresponded to an increase in yeast related OTUs. The main yeast genera were *Candida, Dipodascus*, *Hannaella, Hanseniaspora, Kazachstania*, and *Metschnikowia* (alphabetical order). Throughout this second phase, the relative abundance of the yeasts *Candida* and *Hannaella* increased up to 45% of the total OTUs in LB (22.5 ± 10.7% *Candida*, 21.6 ± 4.7% *Hannaella*) and LB–LH (17.5 ± 7.6% *Candida*, 28.4 ± 20.2% *Hannaella*) treated silages after 32 days. *Kazachstania* was the dominant yeast genus after four (7.0 to 16.6%) and eight (9.4 to 20.7%) days of fermentation, but subsequently decreased in abundance in all but the Control samples. Santos et al. [48] studied the diversity of yeast populations in corn and high moisture corn silages. They reported that isolates obtained on malt agar plates showed that *Candida, Saccharomyces, Pichia,* and *Kazachstania* were the dominant genera. A similar study was performed on corn silage [49], which identified *Issatchenkia, Geotrichum*, and *Pichia* as the dominant genera. Mansfield and Kuldau [50] reported that *Geotrichum candidum* was the dominant yeast in corn silage from Pennsylvania over a two-year period. The differences in key genera between published reports, and also from our study, are likely partly due to the different methods of identification used, as pointed out by Dunière et al. [33] in the case of fungal populations. Santos et al. [13] observed significant discrepancies in populations between samples due to geographic or environmental differences.

PCoA analysis performed from the Bray–Curtis dissimilarity index (Figure 4B) confirmed the trends observed in the heatmap (Figure 5 and Appendix A). The x-axis explained 40.4% of the difference and allowed for a separation of the samples in relation with fermentation time into three groups, the first containing fresh samples, the second containing day 1 and day 2 groups, and the third containing all later ensiling time points (days 4 to 64). Results from a PERMANOVA analysis show that the treatments, opening periods, and the interaction treatments:openings were significant (*p* < 0.001) for the ITS amplicons. The observed variation between samples was related to the inoculation treatment as well as the time of incubation.

After 64 days of fermentation, specific patterns were visible between treatments. The differences were related to the presence of certain yeast genera. An OTU affiliated as *Dipodascus* was present in LB and in LH in nearly 28.3 ± 2.9% and 21.4 ± 1.0%, respectively, of total sequences. Further analysis of the sequence of that OTU using a BLASTn search of GenBank (release 222.0) led to closer identity with the genus *Geotrichum.* Both genera belong to the family *Dipodascaceae* and taxonomic revisions are underway [51], as *Geotrichum* may be the anamorphic state of species of the genera *Dipodascus* and *Galactomyces* [52]. Interestingly, the OTU associated with *Dipodascus* was almost absent from LB–LH and Control samples at day 64 (0.10 ± 0.01% and 0.08 ± 0.01% of the relative abundance), while at day 2, it was present in all four treatments (10.0 ± 2.2% to 13.8 ± 2.8%). The absence of *Dipodascus* OTU in the LB–LH treatment after 64 days of fermentation is hard to explain, since OTUs of this genus are present when LB or LH are used separately. It was not possible to link the absence of *Dipodascus* to the enrichment of another fungal OTU. The yeast *Geotrichum* has already been reported in silage [53] and this genus is also important for the cheese industry, as *Geotrichum candidum* dominates yeast dynamics in some cheeses, particular characteristics of the cheese surface [54], and is considered to be safe by the food industry [55].

The OTU related to *Candida* clearly dominated the Control treatment after 64 days of ensiling, with 50.2 ± 7.5% of the relative abundance. The inoculated LB samples had 9.3 ± 3.1% and 12.9 ± 2.5% for LH (*p* < 0.001) of *Candida* OTUs.

Inoculation with the obligate heterofermentative strains led to significant differences from the Control at day 64. While the Control was dominated by Saccharomycetales (68.4 ± 1.7% relative abundance), LB–LH treated silage had the lowest abundance of OTUs related to that order (22.0 ± 5.1%), and 49.8 ± 1.5% and 47.4 ± 3.0% of relative diversity were observed in LB and LH treated silage samples, respectively (*p* < 0.001 between treatments). At the genus level, this difference was associated with a decrease in *Candida* related OTUs in the inoculated samples. Tremellales were also more abundant in LB–LH treated silage than in the other silages (OTUs of *Hannaella* were the dominant genus in this family), with 41.1 ± 2.1% of the total abundance (*p* = 0.002 versus other treatments). Microbiome analysis of sweet sorghum revealed *Hannaella* early in the fermentation process and was linked to high total sugar content [56].

Sequences affiliated with the yeast genus *Kazachstania* were abundant in the fungal community after 4 and 8 days of fermentation, with highest relative abundance of 20.7 ± 3.4% for the Control at day 8. This population subsequently decreased in all treatments until, by day 32, it represented only a low ratio of the total population, 8.2 ± 2.4% in the Control and 3.2 ± 1.8% of the OTUs in LB–LH (*p =* 0.038). At day 64, the *Kazachstania* OTU ratio increased to a mean of 9.8 ± 1.1% (*p* = 0.222 between treatments).

After 64 days of fermentation, the observed fungal species in LB and LH were greater than in the Control (*p* = 0.022), with 70.0 ± 10.1, 94.3 ± 9.3, 92.3 ± 5.7, and 86.6 ± 6.7 observed species in the Control, LB, LH, and LB–LH treated silages). The presence of the *Dipodascus* OTU was associated with the highest fungal diversity (LB and LH), while *Candida* OTUs were associated with lower fungal alpha diversity (Control). However, care should be taken when extrapolating the results of the contrast analysis (Figure 3) since differences in low-abundance OTUs may be significant. As logCPM values for most OTUs analyzed here were high, we consider that the differences reported between treatments are meaningful.

Although aerobic stability was not measured in our study, the LB–LH combination has been shown to improve aerobic stability in another study using the same forage [57], and several authors demonstrated that *Lb. buchneri* 40788 improves aerobic stability by reducing yeast counts [58]. Thus, we could hypothesize that a better control of yeast population by LB–LH would induce an improved AS in corn silage. Other biochemical mechanisms cannot be ruled out since lower yeast counts may also be associated with a different fungal community profile.

### 3.3. Fermentation Profiling

Changes in pH and in organic acids followed the rapid succession of Eubacteria observed within the first 2 days of fermentation (Figure 6A). The initial drop in pH, down to a mean of 5.01 ± 0.05 (*p* = 0.008) and 4.24 ± 0.01 (*p* = 0.070 between treatments) after 1 and 2 days, respectively, could result from the activity of *Leuconostoc*, *Lactococcus*, and *Enterobacter*, since they collectively represented the dominant OTUs during this period. A change in LAB populations was previously reported and explained by the lower resistance to acid of the primary colonizing species [59]. As the pH reached its lowest value, i.e., a mean of 3.77 ± 0.01 (*p* < 0.001 between treatments) after 8 days of fermentation, the earlier colonizing species were replaced by *Lactobacillaceae*. Although *Leuconostocaceae* species have been reported to be more sensitive to low pH [7], they still represented around 4% of the population in the Control and LB treated silage at the end of the second bacterial population succession, after 16 days of fermentation. Following linear modeling of the pH interactions between time and treatments, the *p* values were all over 0.5, so the four pH curves showed a similar profile (Table 1).

The main difference between treatments was clearly lactic acid and acetic acid concentrations (Table 1, Figure 7B,C). Lactic acid concentration was similar between inoculated samples and the Control up to 16 days of incubation. Subsequently, while the concentration remained at slightly more than 40 g lactic acid·kg^−1^ DM in the Control samples, it dropped in the treated silages. In LB, it dropped to 15.2 ± 0.8 g lactic acid·kg^−1^ DM by day 64 (Figure 6B). The significant drop in lactic acid was not observed in LH samples, even though *Lb. hilgardii* is an obligate heterofermentative LAB [60]. After 64 days of incubation, the lactic acid concentration in the LH treated samples was 35.0 ± 1.2 g·kg^−1^ DM, close to the concentration measured in the Control. At the last opening, the concentration of lactate in the LB–LH treated silages, 23.8 ± 0.8 g·kg^−1^ DM, was between the values measured when LB or LH were inoculated independently. The results of the mixed-effect model (time × treatment) indicated a significant difference in the lactic acid curve in LB compared to the Control, while the difference between LB–LH and the Control was not significant (Table 1). This slight difference characterized the strong similarity in lactate accumulation between LB–LH and LB in several of the seven openings, which mainly differed at day 64 (Figure 6, *p* < 0.001 between treatments). Although LH led to a higher accumulation of lactic acid, changes in the fermentation pattern that occurred early in the process confirm the slight difference in the microbial community structure compared to LB and the LB–LH. As shown by the contrast analysis, LB treated silage harbored a higher diversity of LAB, mainly obligate heterofermentative species, including *Lb. brevis-* and *Lb. ingluviei*-related OTUs. These species could have contributed to the conversion of lactate to acetate in LB [61].

The concentration of acetic acid differed between the three treatments and the Control samples (*p* < 0.001) (Table 1, Figure 6C) during the incubation period, but was within the expected values for corn silage [62]. Differences were already observed after the second day of fermentation, and significant differences were observed at day 8 (*p* < 0.001, SEM = 0.77, with 10.9 ± 1.3 g·kg^−1^ DM for Control and 13.3, 13.8, and 19.8 g·kg^−1^ DM acetic for LB–LH, LB, and LH, respectively). According to the diversity data, Control samples had a higher abundance of unidentified *Lactobacillaceae* and Lactobacillales (Figure 1), which were not necessarily replaced by OTUs affiliated to *Lactobacillus*. After 64 days of incubation, the concentration of acetic acid in LB was 40.4 ± 4.1 g·kg^−1^ DM, similar to that in the LB–LH and LH treatments, with 40.4 ± 4.6 and 36.0 ± 3.1 g·kg^−1^ DM, respectively (*p* < 0.001, SEM = 2.28). The concentration of acetic acid in the Control samples after 64 days of fermentation was 17.5 ± 1.6 g·kg^−1^ DM, i.e., in the range of published reports [5]. In a study of co-inoculation with *Lb. plantarum* and *Lb. buchneri*, Hu et al. [63] observed higher concentrations of acetate using both strains compared to *Lb. buchneri* alone, but the concentration depended on the dry matter content, and overall, no statistical interactions were observed. When comparing the same treatments with a trial on whole plant corn ensiled at 363 g·kg^−1^ dry matter, the concentrations of acetic acid at 64 days correspond to the level reported by 250 days of incubation in Ferrero et al. [57]. The differences between LB–LH and the Control in both studies is similar.

Analyzing the ratio of lactic acid to acetic acid and ethanol concentrations (LA/(AA + EtOH)) (Figure 7A) gave a more precise picture of the fermentation profiles between the homofermentative and heterofermentative lactic acid bacteria populations [64]. In the Control samples, as mentioned earlier, rapid accumulation of lactic acid was observed, triggering an increase in the LA/(AA + EtOH) ratio, up to 2.67 after 32 days, which differed significantly (*p* < 0.001) from the ratios calculated for the inoculated samples, between 1.05 to 1.58, but were in line with published results [65] for homofermentative populations at that period [66] (Table 1). In Control samples, the ratio started to decrease by day 64, indicating a putative succession of the microbiota toward heterofermentative processes. Interestingly, the LA/AA + EtOH ratio for LB–LH was similar to LB at all time points, although the concentrations of both lactic and acetic acid were always higher in LB–LH at the later opening. The higher concentration of lactic acid in LB–LH could delay the increase in pH after opening the silage, and thus help preserve silage for a longer period. The evolution of silage after 64 days of fermentation warrants further investigation to confirm this hypothesis.

We noticed that after 64 days of fermentation, bacterial alpha diversity was 61.5 ± 6.6 species in the Control and decreased to 28.3 ± 2.95 in LB–LH samples (*p* < 0.001) (Appendix A). The strong decrease in the number of observed species in silages inoculated with combined LB and LH could be related to the accumulation of organic acids, in particular of lactic and acetic acid whose concentrations increased in those samples. At low pH, an increase in acetic acid could create a stressful environment for most microorganisms, as the higher pK_a_ of acetic acid (4.76) compared to the pH of the silage allows nearly all acetate to be in its undissociated form [67]. Acetic acid could then easily enter microbial cells, dissociate, and decrease the intracellular pH associated with notable energy expenditure to maintain physiological pH values, eventually killing cells unable to adapt. Although the Control silages reached a pH comparable to that of the treated samples, they had a lower concentration of acetic acid at 64 days of fermentation. This pH would be too low to cause high bacterial and fungal mortality, but as its concentration progressively increased during the course of fermentation in the Control, acetic acid could induce later changes in the microbial population. Our results confirm this hypothesis, since alpha diversity in the Control samples increased from a low of 47.7 ± 3.5 observed species at day 16, while, in the inoculation treatments, it decreased after that period. Therefore, heterofermentative bacterial inoculants offer an effective solution to ensure high levels of acetic acid in silage, thereby improving the stability of the silage.

The synthesis of propionic acid follows a specific pathway, putatively involving other members of the lactic acid bacterial population, i.e., *Lactobacillus diolivorans* [68] or *Lactobacillus reuteri* [69]. A complex of 22 enzymes has been shown to be linked to the degradation of 1,2-propanediol to non-equimolar levels of 1-propanol (reduced) or propionic acid (oxidized), with propionaldehyde as an intermediate in the reaction [70,71]. In the Control silages, concentrations of these three compounds remained below detection levels throughout the 64-day period of the trial (Figure 7).

Conversely, following inoculation, the concentration of 1,2-propanediol increased irrespective of the strain. After 64 days of fermentation, the concentration of 1,2-propanediol continued to increase in LH and LB–LH treated silages, whereas it started to decrease in LB, leading to a significant (*p* < 0.001) difference between treatments at this time. The lower concentration of 1,2-propanediol with LB was possibly due to transformation into propionic acid, whose concentration concomitantly increased (Figure 7D). The production of 1-propanol (Figure 7C) and propionate (Figure 7D) were mostly influenced by the presence of *Lb. buchneri*, and treatments with *Lb. hilgardii* had only a minor influence on this metabolism. The LB–LH treatment produced an intermediate or delayed response. The concentration of propionic acid in LB–LH was only about half that obtained in LB alone after 64 days of fermentation. It is hard to explain why the concentration of propionic acid does not follow the same trend in LB–LH and in LB for both compounds. It can be hypothesized that the transformation of 1,2-propanediol into propionic acid is more efficient in LB following stimulation of specific microbial populations [70,72], a process that occurs to a lesser extent in the LB–LH treated silage.

## 4. Conclusions

Using amplicon metasequencing analysis of the microbial communities over the course of fermentation revealed three distinct community profiles from the 16S amplicons, and two from the ITS amplicons. Differences in lactate and acetate concentrations between treated silages could influence both bacterial and fungal communities, and after 64 days of fermentation, the LB–LH treated silages harbored a distinctive yeast community, with *Hanaella* as the main OTU. In the separate LB and LH inoculated silages, *Diplodascus* occupied this niche, and *Candida* dominated the Control samples. LB–LH also had the lowest alpha diversity level of bacteria, which influenced both bacterial and fungal communities to improve forage preservation and potentially animal performance. The LB–LH combination brings specific changes that are distinct to the sum of their individual impact on the microbial communities.

Significant differences in fermentation profiles were associated with each inoculant, and changes in fungal communities will need further investigation to understand the mechanisms of action of the bacterial inoculants, especially the combination LB–LH.

## Figures and Tables

**Figure 1 microorganisms-07-00595-f001:**
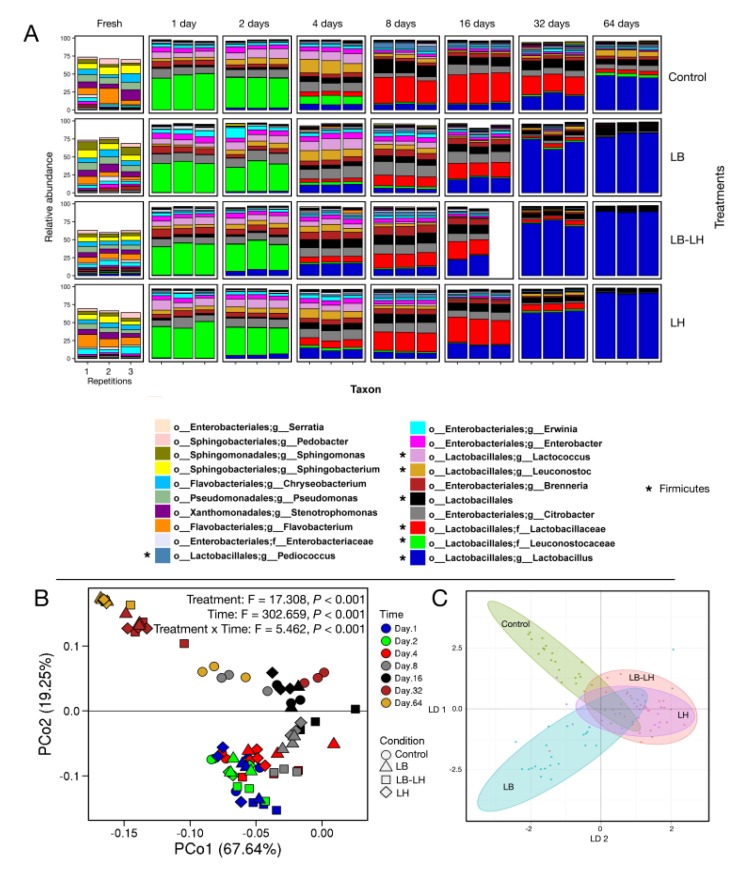
Bacterial composition by 16S rDNA gene amplicon metasequencing (V4 region) in corn silages under four inoculation treatments for eight fermentation periods, between fresh and 64 days of ensiling (**A**). Each panel corresponds to one fermentation treatment (inoculant × time) and the diversity results of three independent repetitions. The color code shows the 20 most abundant taxa computed using all 32 ensiling conditions. Taxon identification was at the genus level. Results of the principal coordinates analysis (**B**) and the linear discriminant analysis (**C**) computed from the complete metasequencing results are shown. Principal component analysis of the weighted Unifrac distance values was performed without the results for the fresh samples because the dissimilarity was so great that the effect of fermentation time was hard to visualize. Operational taxonomic units (OTUs) identified by * belong to the Firmicutes phylum.

**Figure 2 microorganisms-07-00595-f002:**
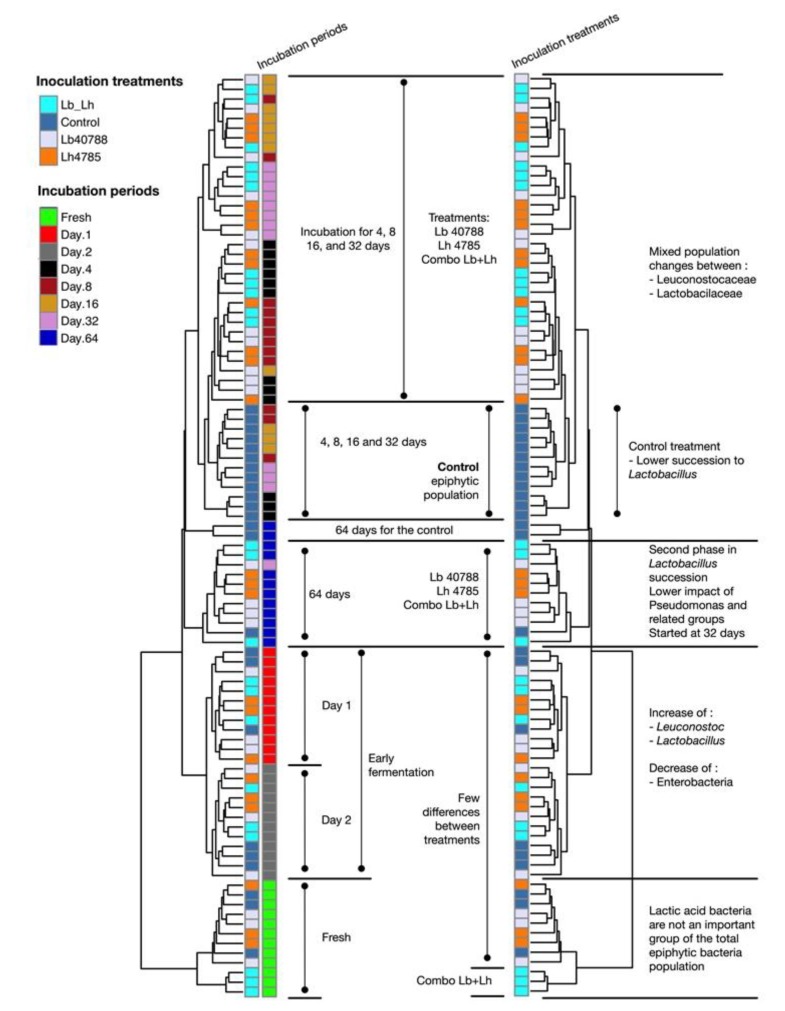
Main variations in Eubacteria diversity using the UPGMA dendrogram generated for analysis of the heatmap (see Appendix A).

**Figure 3 microorganisms-07-00595-f003:**
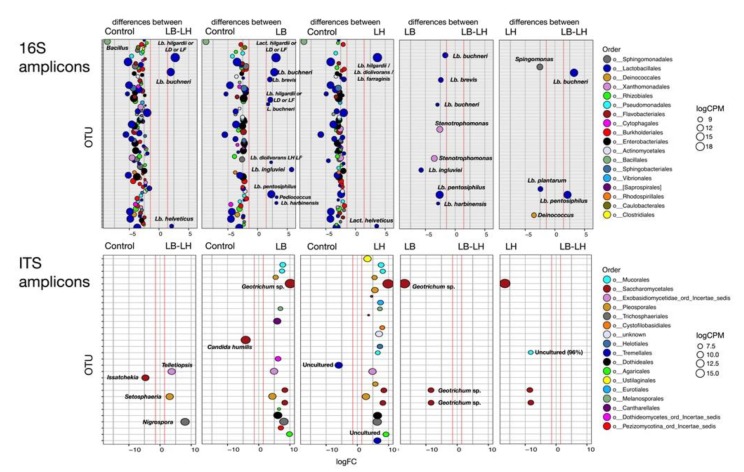
Contrast analysis of amplicon sequencing data of the samples after 64 days of ensiling using the EdgeR package from the 16S amplicons and the ITS amplicons. OTUs whose abundance differed significantly between the Control and the three inoculation treatments or between the inoculation treatments themselves are shown. LogFC (fold changes) and logCPM (count per million) were used for the representation. Only OTUs with a logFC ≤1.5 (red line in each panel) and false discovery rate (FDR) value <0.05 were considered as being differentially abundant. Color coding is based on the taxonomic classification of individual OTUs. Taxonomic identification of the OTU affiliated as *Lb. hilgardii* could also be related to *Lb. diolivorans* or *Lb. farraginis*, according to the current database search (LH-LD-LF).

**Figure 4 microorganisms-07-00595-f004:**
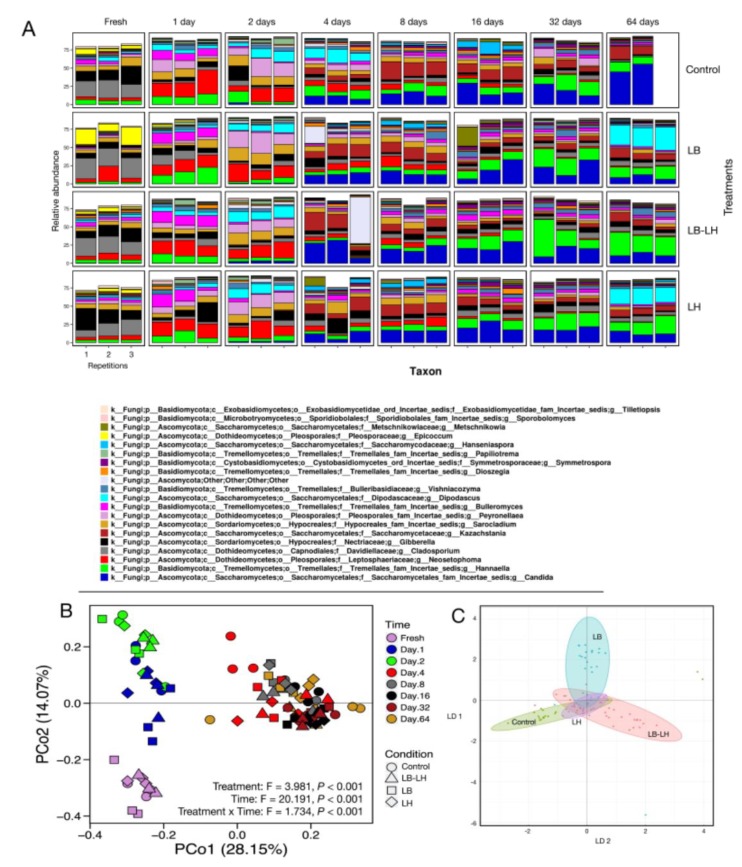
Fungal composition according to ITS amplicon metasequencing (ITS1 region) from corn silages under four inoculation treatments and eight fermentation periods, between fresh and 64 days of ensiling (**A**). Each panel corresponds to a fermentation condition (inoculation treatment × time) and the sequencing results of three independent repetitions. The color coding represents the 20 most abundant taxa, computed using all 32 ensiling conditions. Taxon identification was at the genus level. Principal component analysis (**B**) and linear discriminant analysis (**C**) computed from the complete metasequencing results are presented. The PCA was performed from the Bray–Curtis dissimilarity matrix data.

**Figure 5 microorganisms-07-00595-f005:**
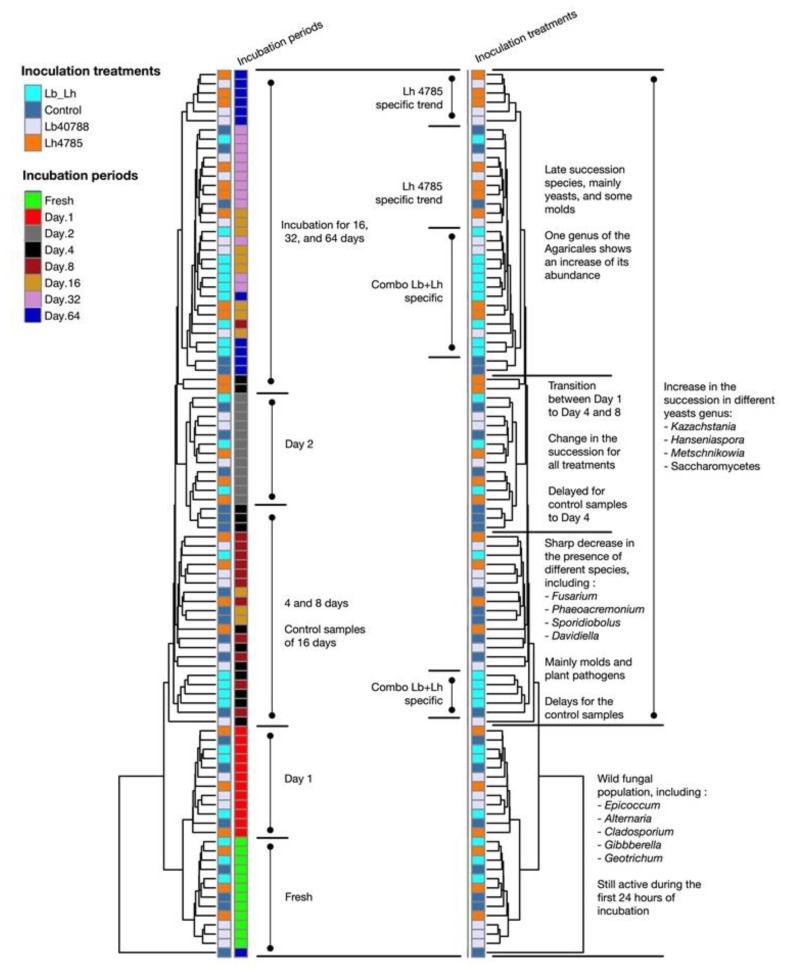
Main changes in fungal diversity using the UPGMA dendrogram generated for analysis of the heatmap (see Appendix A).

**Figure 6 microorganisms-07-00595-f006:**
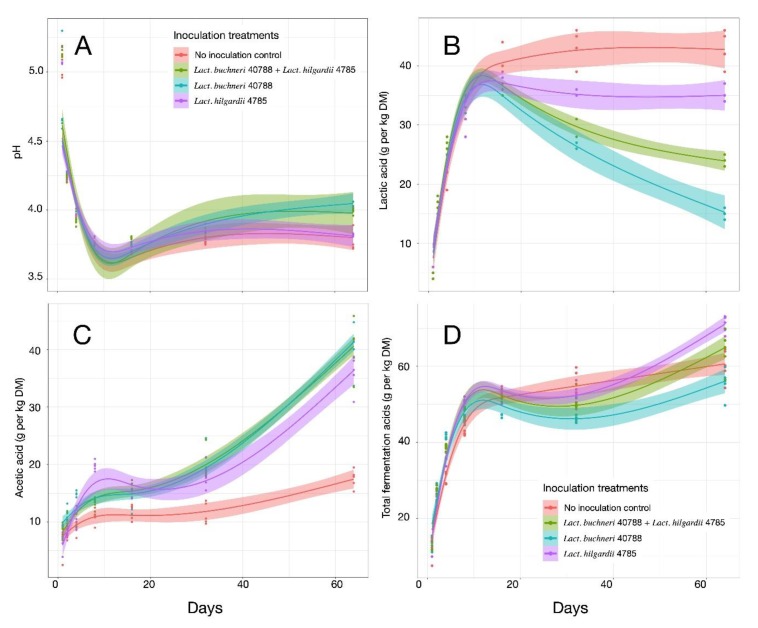
Fermentation kinetics of independent samples of corn silages harvested after 1, 2, 4, 8, 16, 32, and 64 days of fermentation. The following four inoculation treatments were applied: Control, *Lb. buchneri*, *Lb. hilgardii*, and a combination of the two LAB species. The four panels show pH (**A**), the concentration of lactic acid (**B**), the concentration of acetic acid (**C**), and the concentration of total volatile fatty acids (**D**). The trend lines correspond to a smoothing function computed by linear modelling using the y~ns(x,3) equation. The shaded area boarding each trend line represents the 95% confidence interval.

**Figure 7 microorganisms-07-00595-f007:**
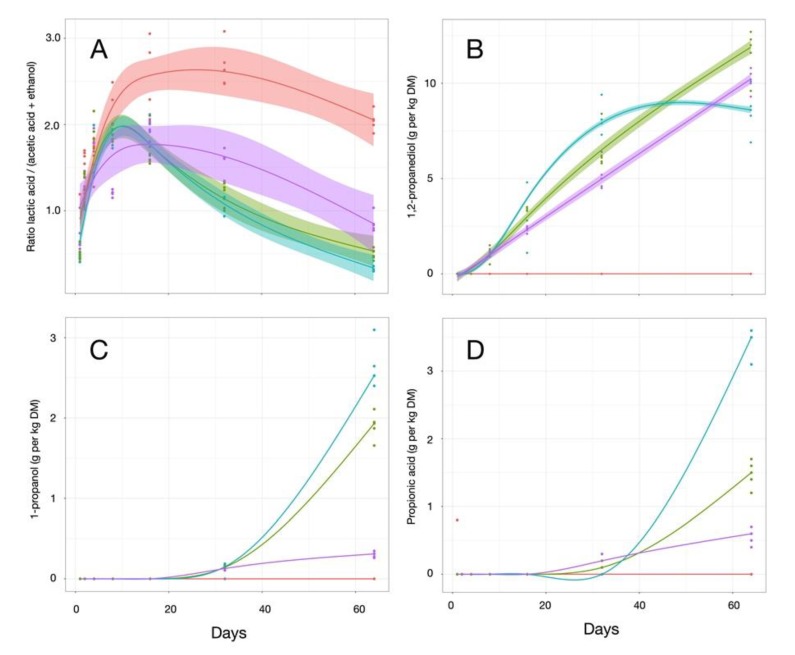
Ratio of the concentration of lactic acid/(acetic acid + ethanol) (**A**) and propionate metabolic pathway compounds (1,2-propanediol (**B**), 1-propanol (**C**), propionic acid (**D**)) from independent samples of corn silage harvested after 1, 2, 4, 8, 16, 32, and 64 days of fermentation. The following four inoculation treatments were applied: Control, *Lb. buchneri*, *Lb. hilgardii*, and a combination of the two LAB species. The trend lines correspond to a smoothing function computed by linear modelling using the y~ns(x,3) equation. The shaded area boarding each trend line represents the 95% confidence interval.

**Table 1 microorganisms-07-00595-t001:** Results of the mixed effect model analysis for comparison of the fermentation profiles at all opening periods between the control and the bacterial inoculation treatments.

Treatments	pH ^a^	Total VFA ^a^	Lactic Acid ^a^	Acetic Acid ^a^	Ratio Lactate/(Acetate + EtOH) ^a^
Lact. buchneri 40788	0.855	0.972	0.011	<0.001	<0.001
Lact. hilgardii 4785	0.736	0.206	0.410	<0.001	<0.001
Combo LB–LH	0.552	0.396	0.109	<0.001	<0.001

^a^*Probability* of a difference between the bacterial inoculation and the control treatments. VFA, volatile fatty acids.

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
