# Peer review of "Dynamic Succession of Microbiota during Ensiling of Whole Plant Corn Following Inoculation with Lactobacillus buchneri and Lactobacillus hilgardii Alone or in Combination"

_microorganisms, 2019, doi:10.3390/microorganisms7120595_

Round 1

Reviewer 1 Report

The authors present a well-designed study that investigates the impact of Lactobacillus strains on the microbial composition in mini-silos sampled over ca. two months. More precisely, the authors compare the effect of three treatments (L. buchneri, L. higardii and both) with the control (water) on bacteria and fungi. The authors clearly demonstrate an effect of Lactobacillus addition on microbial community composition.

Do microbial loads differ for the different treatments?

Did the temperature change in the mini-silos over time?

What drives the observed succession? The pH slowly increases after 10 days, without bringing back the taxa that previously thrived. Which additional mechanisms beyond acidification could be at play? A triplot (e.g. a biplot combined with envfit in vegan) or a correlation analysis may give an idea of which taxa covary with which metabolites.

The manuscript contains lengthy descriptions of the successional stages. These could be summarised in a table, which could also list how these stages compare to observations in other studies.

What happened with the third replicate in the bacterial LB-LH treatment for day 16 and the third replicate in the fungal control in day 64?

L. 92-93: "A total of 160 vacuum-bag mini-silos were prepared, 40 per treatment with five repetitions for each treatment × time factors"

There are only three replicates and not five shown in the figures, so 24 per treatment.

L. 160: "Alpha diversity (observed OTUs) was computed from un-normalized OTU table for both 16S and ITS"

How do authors deal with differences in sequencing depth that bias alpha diversity? For alpha diversity computation, rarefaction is a straightforward way to handle diversity differences due to varying sequencing depth.

L. 161-163: "For 16S data, weighted Unifrac distances were generated using the normalized OTU table and phylogenetic tree previously generated. For ITS data, Bray-Curtis dissimilarity metric was generated from the normalized OTU table."

Why were two different measures of dissimilarity used for 16S and ITS data?

L. 209: Please replace "corn" by "maize", since the term is ambiguous (including also cereal crops).

L. 226-228: In line 220 the authors say that in 24-48 hours a shift occurred and then state that these results are in line with timepoint 0 of the cited paper. Do they mean that their results from timepoint 24-48 are the same as timepoint 0 of the cited paper, or is the timepoint 0 of this manuscript in line with timepoint 0 of the cited paper?

Figure 1A: What is the role of the asterisk in Figure 1A? If the point is to highlight Lactobacillales, the asterisk text should be changed.

L. 59: extremely effective -> effective

L. 191: linear discriminant assay -> linear discriminant analysis

L. 316: the sequenced -> the sequences

L. 475: The letter size is too small

Author Response

The authors present a well-designed study that investigates the impact of Lactobacillus strains on the microbial composition in mini-silos sampled over ca. two months. More precisely, the authors compare the effect of three treatments (L. buchneri, L. higardii and both) with the control (water) on bacteria and fungi. The authors clearly demonstrate an effect of Lactobacillus addition on microbial community composition.

Do microbial loads differ for the different treatments?

A: Inoculation rate is the same between the different inoculation treatments, with 4 x 105 CFU per g of fresh matter. The negative control was not inoculated, so initial loads do differ on this account. With a mean count of log10 5.81 (6.48 x105 cfu/g), this means that the load was in the same range for all four treatments.

Initial load was added to the text.

Did the temperature change in the mini-silos over time?

A: This is effectively an important question. Although the mini-silos were disposed according to a randomized pattern, we took care to have enough space in between them to allow good ventilation during incubation. Considering the amount of forage inside individual vacuum bag, changes in temperature would be minimal. The correspond section in the methodology section was updated (first paragraph) accordingly.

What drives the observed succession? The pH slowly increases after 10 days, without bringing back the taxa that previously thrived. Which additional mechanisms beyond acidification could be at play? A triplot (e.g. a biplot combined with envfit in vegan) or a correlation analysis may give an idea of which taxa covary with which metabolites.

A: The data presented in this manuscript will soon be accompaign by a follow-up. We did full metasequencing analysis from some of the samples (day 2, 8, and 64) and also metabolomic analysis. The data is currently being analyzed and this will allow to perform the proposed correlation between microbiome-metabolome-wet chemistry.

The manuscript contains lengthy descriptions of the successional stages. These could be summarised in a table, which could also list how these stages compare to observations in other studies.

A: We tried to summarize the different stages of succession of the microbiota, from both the 16S and ITS amplicons in Figure 2 and Figure 5. Adding a table to summarize this information would duplicate the function of these two figures.

The inner section of the figures points out the time period and the corresponding impact on specific treatments. The right side explain in some details the observed changes in microbiota.

We can effectively update the figures with observations from other studies, but this will reduce the ease of reading the figures. Although some recent articles performed similar successional analysis, their used of different inoculants or time periods would be hard to match.

Please advise us of your intend in relation with replacing the figures with tables.

What happened with the third replicate in the bacterial LB-LH treatment for day 16 and the third replicate in the fungal control in day 64?

A: The failed sample for Lb-Lh day 16 condition (for 16S data) only gave 45 reads and was therefore left out. The failed sample for Control day 64 (ITS data) only gave 6 fungal read (the rest were contaminants) and was also left out. These details were added at lines 181-184.

L. 92-93: "A total of 160 vacuum-bag mini-silos were prepared, 40 per treatment with five repetitions for each treatment × time factors"

There are only three replicates and not five shown in the figures, so 24 per treatment.

A: All biochemical analysis was performed on all five repetitions, but DNA extraction and sequencing were performed on three repetitions.  Section 2.3 was updated accordingly. Caption of the figures report to the correct information.

L. 160: "Alpha diversity (observed OTUs) was computed from un-normalized OTU table for both 16S and ITS"

How do authors deal with differences in sequencing depth that bias alpha diversity? For alpha diversity computation, rarefaction is a straightforward way to handle diversity differences due to varying sequencing depth.

A: We thanks the reviewer for pointing this issue. After a second reading, we realized that we lacked clarity on that statement. To obtain alpha diversity values, un-normalized OTU tables were rarefied at 1,000 reads and alpha-diversity values were computed from these rarefied tables. This has been updated in the text (lines 175-177).

L. 161-163: "For 16S data, weighted Unifrac distances were generated using the normalized OTU table and phylogenetic tree previously generated. For ITS data, Bray-Curtis dissimilarity metric was generated from the normalized OTU table."

Why were two different measures of dissimilarity used for 16S and ITS data?
A: Using a phylogenetic tree for fungi community analysis is controversial. Internal validations suggest that global alignments, inherent to generating the UniFrac distance matrix, may be impractical because of the high variability in sequence and length of ITS amplicons. This problematic was also highlighted by Lindahl et al., New Phytol. (2013);199:288-99.

L. 209: Please replace "corn" by "maize", since the term is ambiguous (including also cereal crops).

A: Done

L. 226-228: In line 220 the authors say that in 24-48 hours a shift occurred and then state that these results are in line with timepoint 0 of the cited paper. Do they mean that their results from timepoint 24-48 are the same as timepoint 0 of the cited paper, or is the timepoint 0 of this manuscript in line with timepoint 0 of the cited paper?

A: The sentence had been improved by referring directly to the time period we are referring to for this comparison (24h).

Figure 1A: What is the role of the asterisk in Figure 1A? If the point is to highlight Lactobacillales, the asterisk text should be changed.

A: The asterisk is to highlight which of the OTUs are of the Firmicutes phylum. The figure already has this information (above the legend of 1A) but will copy to the caption also.

L. 59: extremely effective -> effective

A: Done

L. 191: linear discriminant assay -> linear discriminant analysis

A: Done

L. 316: the sequenced -> the sequences
A: Done

L. 475: The letter size is too small

A: Corrected in the manuscript

Reviewer 2 Report

General comments:

This paper reveals the bacterial and fungal community and the corresponding fermentation parameter changes during the process of ensiling. Since corn silage is a commonly used feed in the US and globally, the information provided by this paper is valuable. However, a lot of the information described in the text cannot be easily matched with the figures. Therefore, major revision is recommended to improve the quality of the manuscript and the writing of the entire manuscript should be checked by a native English speaker.

Specific comments:

Abstract

Line 19-20: Bacterial genus should be written as L. buchneri, not Lb. buchneri. This also apply to other bacteria names and should be corrected through the whole manuscript. In addition, corn silage should be mentioned in this sentence

Line 27: the sentence “which also led to specific yeast OTUs” is not complete and meaningful, please rephrase

Introduction:

Difference between homofermentative and heterofermentative lactic acid bacteria should be given in the introduction as the fermentation profiles are discussed extensively in the results and discussion section

Line 41: change “colonies” to “isolates”

Line 63: It is stated that “few microbial diversity studies …”. The major findings from these few studies should be introduced briefly and cited

Materials and methods:

Line 99: delete “also”

Line100: diluting 25 g of sample in 200 mL of water is not a 1:10 dilution. Please explain.

Line 101: Why were the samples incubated under refrigeration overnight? During this overnight incubation, the microorganisms may continue producing VFAs and change the fermentation profiles.

Results and discussion:

Supplementary data should be organized as the order they appear in the text

In the whole results section, the use of Lactobacillus and Lactobacillaceae are very confusing. Since Lactobacillus belongs to Lactobacillaceae, it is hard to understand when mentioning Lactobacillaceae, if Lactobacillus is included or not. Especially in Fig 1, they were separated from each other. Clarification needs to be given.

Line 217: Since “… in different forage species” is mentioned, more than one citation should be given to represent “different” species.

Line 228: the paragraph is majorly talking about the microbial community change in 24-48 h, but there is no comparison of the current study with other studies about this period. Only comparison of 0 h is described.

Line 247-249: the results cannot be visualized by reading Figure 1. This also applies to the extensive discussion of Lactobacillus and Lactobacillaceae abundance described in the following paragraphs. Making a separate figure with bar graphs or growth curves with the most important bacterial and fungal taxa (e.g.  Lactobacillus, Lactobacillaceae, Candidas, and Kazachstania) is highly recommended.  

Line 319: table numbers need to be provided

Line 328: Please explain “Lactobacillales were more abundant in Control”. From the figure, it looks like Lactobacillales were more abundant in inoculated treatments.

Line 397: The number on X-axis was 40.41%, not 28.2%? Please explain.

Line 475: Font size is different

Line 488: The P-value was 0.109, which is higher than 0.1. Why is it called a trend? And since LB vs control was significant whereas LB-LH vs. control was not significant, the difference between LB and LB-LH should not be called a “slight difference” and there is no “strong similarity”

Line 496: Have the abilities of LB or LH to produce acetic acid been studied previously? If yes, the quantity of acetic acid produced in previous studies should be discussed and compared to the results of this study respectively.

Line 521: What LA/AA+EtOH indicates and why it is important for silage fermentation should be explained before using this parameter.

Line 531: add “after opening silage” after “…in pH”

Line 539-541: citation needs to be added.

Conclusions:

A more concise conclusion should be given to summarize the major findings.

Figures

Fig. 1A: the word “Treatments” should be on the right side of the figure as the actual treatments are given there. The asterisks are not aligned well with the text (e.g. * and Firmicutes are not in the same line).

Fig. 1B: the number given at X (67.64%) and Y (19.25%) axis should be explained in the figure legends.

The problems in Fig 1 also applies to figure 4

Figure 2 & 5: supplementary figure needs to be mentioned

Figure 3: subtitles A,B,C,… should be given to each subfigure and used in the text so the readers can understand easier what is referring to.

Figure 6: Figure captions are wrong. A,B,C,D do not match with the parameters measured; The four colors used in the figure (also figure 7) are too similar to each other, making it hard to differentiate the treatments, especially for individual sample point. Changing to colors with better contrasts is highly recommended.  

Figure S1&2: the heat map color key is not explained.

Figure S3: there should be no “treatment” in the figure

Figure S4: Subtitles ABCD are not explained in the figure captions

Author Response

General comments:

This paper reveals the bacterial and fungal community and the corresponding fermentation parameter changes during the process of ensiling. Since corn silage is a commonly used feed in the US and globally, the information provided by this paper is valuable. However, a lot of the information described in the text cannot be easily matched with the figures. Therefore, major revision is recommended to improve the quality of the manuscript and the writing of the entire manuscript should be checked by a native English speaker.

Specific comments: Abstract

Line 19-20: Bacterial genus should be written as L. buchneri, not Lb. buchneri. This also apply to other bacteria names and should be corrected through the whole manuscript. In addition, corn silage should be mentioned in this sentence

A: The sentence had been changed and now starts with “Corn silage” instead of only “Silage”.

Since the manuscript refer also to other genus within the Lactobaciliales, Lactococcus as an example, the two letters abbreviation official nomenclature was used in the text instead of the one letter abbreviation. Please refer to Holzapfel W. H. & Wood B. J. B. Lactic Acid Bacteria: biodiversity and taxonomy. John Wiley & Sons. 2014 for details.

Line 27: the sentence “which also led to specific yeast OTUs” is not complete and meaningful, please rephrase

A: The sentence was update with additional specification relating to the observed diversity profile.

Introduction:

Difference between homofermentative and heterofermentative lactic acid bacteria should be given in the introduction as the fermentation profiles are discussed extensively in the results and discussion section

A: Third paragraph modified to better define the homofermentative and heterofermentative LAB (updated lines 55-59).

Line 41: change “colonies” to “isolates”

A: Done

Comments and Suggestions for Authors

Line 63: It is stated that “few microbial diversity studies ...”. The major findings from these few studies should be introduced briefly and cited

A: Results from these studies are further discussed at the beginning of the results & discussion section and includes some citation to these studies.

Materials and methods:

Line 99: delete “also”

A: Done

Line100: diluting 25 g of sample in 200 mL of water is not a 1:10 dilution. Please explain.

A: We did not mention that this was a 1:10 dilution. The text refers directly to the protocol provided by the analytical lab. Please see “https://www.foragelab.com/Resources/Lab-Procedures” as the source of the information.

Line 101: Why were the samples incubated under refrigeration overnight? During this overnight incubation, the microorganisms may continue producing VFAs and change the fermentation profiles.

A: Idem as previous reply. Please see “https://www.foragelab.com/Resources/Lab-Procedures” as the source of the information.

The laboratory was contacted for further details. They replied that this period had not impact on the concentration of the different VFAs.

Our normal protocol in the lab while doing VFA analysis is to let the extract settle for 2 hours.

Results and discussion:

Supplementary data should be organized as the order they appear in the text

A: Done

In the whole results section, the use of Lactobacillus and Lactobacillaceae are very confusing. Since Lactobacillus belongs to Lactobacillaceae, it is hard to understand when mentioning Lactobacillaceae, if Lactobacillus is included or not. Especially in Fig 1, they were separated from each other. Clarification needs to be given.

A: We understand the reviewer’s concern and we acknowledge that discussing these two lineages can be confusing. The reason is that some OTUs were confidently assigned a lineage to up to the genus level (i.e. Lactobacillus) while others could only make it to the family level (Lactobacillaceae). In consequence, when we compute our taxonomic summaries from our OTU tables, we necessarily end up with two different lineages : Lactobacillaceae;Lactobacillus and Lactobacillaceae. On way we could have come around this would have been to perform taxonomic summaries at the family level, but we feel that we would have lost important information on the OTUs whose classification made it to up to the genus level. We tried to re-word as much as possible to avoid confusions. 

Line 217: Since “... in different forage species” is mentioned, more than one citation should be given to represent “different” species.

A: It would be possible to add the following article with the one at line 241. We did not add directly to the manuscript since we did not want to change the coding in relation with the citation and bibliography section.

Bolsen, K. K., C. Lin, B. E. Brent, A. M. Feyerherm, J. E. Urban, and W. R. Aimutis. “Effect of Silage Additives on the Microbial Succession and Fermentation Process of Alfalfa and Corn Silages.” Journal of Dairy Science 75 (1992): 3066–83.

Line 228: the paragraph is majorly talking about the microbial community change in 24-48 h, but there is no comparison of the current study with other studies about this period. Only comparison of 0 h is described.

A: A new citation was added to this section. We already used this citation in another section of the manuscript.

Line 247-249: the results cannot be visualized by reading Figure 1. This also applies to the extensive discussion of Lactobacillus and Lactobacillaceae abundance described in the following paragraphs. Making a separate figure with bar graphs or growth curves with the most important bacterial and fungal taxa (e.g. Lactobacillus, Lactobacillaceae, Candidas, and Kazachstania) is highly recommended.

A: We do not fully understand the proposition of the reviewer. Since the change in microbial diversity corresponds increase of the different microorganisms on or within plant cells on the harvested forage, and several OTUs do not corresponds to isolated strains, production of growth curves will be quite complex.

We do have growth curves results in the laboratory, performed on strains grown on corn infusion. We fear that the stresses encountered by the lactic acid bacteria strain and complexity of that specific environment could hardly be simulated by in vitro studies.

Line 319: table numbers need to be provided

A: Done

Line 328: Please explain “Lactobacillales were more abundant in Control”. From the figure, it looks like Lactobacillales were more abundant in inoculated treatments.

A: The statistical analysis was performed from the numerical data set from different level of phylogenic identification (Genus, Family, Order, or Phylum). In this case from the text, the analysis was performed from the results at the Order level, so direct visual link to Figure 1 (or 4) is not possible.

Please also refer to our answer to the second comments on this section (Results and Discussion) for a precision about identification of some OTUs id as Lactobaccillales.

Line 397: The number on X-axis was 40.41%, not 28.2%? Please explain.

A: We updated the number in the text to reflect the results on the figure. The value of 28.2% was from a previous version of the analysis using a different normalized data set.

Line 475: Font size is different

A: Corrected in the manuscript

Line 488: The P-value was 0.109, which is higher than 0.1. Why is it called a trend? And since LB vs control was significant whereas LB-LH vs. control was not significant, the difference between LB and LB- LH should not be called a “slight difference” and there is no “strong similarity”

A: The text was change by replacing “only a trend” by “not significant”. For the following part of the comments, individual ANOVA were performed for each time point also (data not shown or discussed in the text in this version of the manuscript – it was present in an earlier version and was removed to allow an easier to read discussion. The manuscript was updated at line 537-540 with “(data not shown following ANOVA performed at each time periods)”

Line 496: Have the abilities of LB or LH to produce acetic acid been studied previously? If yes, the quantity of acetic acid produced in previous studies should be discussed and compared to the results of this study respectively.

A: Yes, this is a general mechanism of the obligate heterofermentative LAB. The concentration observed in this study falls within normal range. A new citation was added.

Kung, L. Jr, R. D. Shaver, R. J. Grant, and R. J. Schmidt. “Silage Review: Interpretation of Chemical, Microbial, and Organoleptic Components of Silages.” Journal of Dairy Science 101 (2018): 4020–33. https://doi.org/10.3168/jds.2017-13909.

Line 521: What LA/AA+EtOH indicates and why it is important for silage fermentation should be explained before using this parameter.

A: Thank you for asking for this specification in relation with the ratio of lactic acid of acetic acid + ethanol. I tried to change the sentence to include more details, but it just destroys the paragraph. The citation included at the end of the sentence had the function to lead readers to additional details.

The general ratio used in silage analysis is mainly LA/AA. Adding EtOH allows for the inclusion of the production of EtOH with lactic acid bacteria are using the pyruvate formate lyase and pyruvate dehydrogenase pathways.

The manuscript was not change.

Line 531: add “after opening silage” after “...in pH” Line 539-541: citation needs to be added.

A: Done

For the editor, please add:

Russell, J. B. and F. Diez-Gonzalez. 1997. The effects of fermentation acids on bacterial growth. Advances in Microbial Physiology. 39:205-234.

Conclusions:

A more concise conclusion should be given to summarize the major findings.

A: We did remove some information for the conclusion. It is now more concise and closely linked to the main findings.

Figures

Fig. 1A: the word “Treatments” should be on the right side of the figure as the actual treatments are given there. The asterisks are not aligned well with the text (e.g. * and Firmicutes are not in the same line).

A: Thank you for this observation. An update version of Figures 1 and 4 were prepared. For the alignment problem with the *, there was a change of font between my figure and the one composed by the journal. I will provide the updated .jpg version and will see how this will change in the final version. The * had been moved to a better placement in the figure.

Fig. 1B: the number given at X (67.64%) and Y (19.25%) axis should be explained in the figure legends.

The problems in Fig 1 also applies to figure 4

A: The percentages values reflects the amount proportion of variabilty that is explained by each axe. In our case, the vast majority of variability in our samples is largely explained by both X and Y axes (67.64% + 19.25% = 86.89%). This was included in the figure caption in the revised manuscript.

Figure 2 & 5: supplementary figure needs to be mentioned

A: Done

Figure 3: subtitles A,B,C,... should be given to each subfigure and used in the text so the readers can understand easier what is referring to.

A: The caption explains general analytical explanation on the figure. The first sentence was updated to mention that the data covered both 16S and ITS amplicons.

Figure 6: Figure captions are wrong. A,B,C,D do not match with the parameters measured; The four colors used in the figure (also figure 7) are too similar to each other, making it hard to differentiate the treatments, especially for individual sample point. Changing to colors with better contrasts is highly recommended.

A: Thank you for picking this up. This effectively had to be update. The order of the panels was changed to reflect the text, but the caption was not updated accordingly.

Done.

Figure S1&2: the heat map color key is not explained.

A: Will provide an update version of the figure. The color key was already included in the figure. It was updated with a legend corresponding to the scale (log2(CPM). The caption have been updated also.

Figure S3: there should be no “treatment” in the figure
A: An updated version of the figure was prepared. Thank you for this comment.

Figure S4: Subtitles ABCD are not explained in the figure captions

A: The ABCD subtitles had been removed and were replaced by updated title of each panel.

Round 2

Reviewer 2 Report

The corrected figures are not shown in the updated manuscript, but only described in the response to reviewers. Therefore, they cannot be reviewed easily. Other than that, the paper is good to be accepted in present form.